# Why the Indo-Gangetic Plain is the region with the largest NH₃ column in the globe during pre-monsoon and monsoon seasons?

Tiantian Wang[1], Yu Song[1], Zhenying Xu[1], Mingxu Liu[1], Tingting Xu[1,2], Wenling Liao[1], Lifei Yin[1], Xuhui Cai[1], Ling Kang[1], Hongsheng Zhang[3], Tong Zhu[1,4]

[1]State Key Joint Laboratory of Environmental Simulation and Pollution Control, Department of Environmental Science, Peking University, Beijing 100871, China
[2]Environmental College, Chengdu University of Technology, Chengdu 610059, China
[3]Laboratory for Climate and Ocean-Atmosphere Studies, Department of Atmospheric and Oceanic Sciences, School of Physics, Peking University, Beijing 100871, China
[4]Beijing Innovation Center for Engineering Science and Advance Technology, Peking University, Beijing 100871, China

*Correspondence to*: Yu Song (songyu@pku.edu.cn)

**Abstract.** Satellite observations show a global maximum in ammonia (NH₃) over the Indo-Gangetic Plain (IGP), with a peak from June to August. However, it has never been explained explicitly. In this study, we investigated the causes of high NH₃ loading over the IGP during pre-monsoon and monsoon seasons using WRF-Chem (Weather Research and Forecasting model coupled to chemistry). IGP has relatively high NH₃ emission fluxes (0.4 t km$^{-2}$ month$^{-1}$) due to intensive agricultural activities and high air temperature from June to August. Additionally, low sulfur dioxide (SO₂) and nitrogen oxides (NOₓ) emissions and high air temperature limit the gas-to-particle conversion of NH₃, particularly for ammonium nitrate formation. Moreover, the barrier effects of the Himalayas in combination with the surface convergence weaken the horizontal diffusion of NH₃. The high NH₃ loading over the IGP mainly results from the low gas-to-particle partitioning of NH₃ caused by low SO₂ and NOₓ emissions. It contrasts to those in the North China Plain, where high SO₂ and NOₓ emissions promote the conversion of gaseous NH₃ into particulate ammonium.

## 1 Introduction

Ammonia (NH₃) has multiple environmental implications. As the only alkaline gas in the atmosphere, it reacts with sulfuric acid (H₂SO₄) or nitric acid (HNO₃) to produce ammonium (NH₄⁺) containing aerosols (Seinfeld and Pandis, 2006), which can affect Earth's radiative balance (Abbatt et al., 2006;Adams et al., 2001) and endanger public health (Pope et al., 2002;Stokstad, 2014). In addition, NH₃ is the main form of reactive nitrogen in the environment (Reis et al., 2009), the deposition of ammonia and ammonium can cause acidification of terrestrial ecosystems and eutrophication of water bodies (Paerl et al., 2014).

Satellite observations (Van Damme et al., 2018;Warner et al., 2016) and ground-based measurements (Carmichael et al., 2003) have revealed that the Indo-Gangetic Plain (IGP) has the global maximum NH₃ loading, particularly from June to August. Previous studies have suggested that the high NH₃ loading over the IGP is caused by high NH₃ emissions from intensive agricultural activities(Clarisse et al., 2009;Van Damme et al., 2015b). Interestingly, satellite measurements show that the total columns of NH₃ over the IGP are much higher than those over the North China Plain (NCP), which has higher

NH$_3$ emissions fluxes (www.meicmodel.org/dataset-mix). Therefore, emissions alone might not be enough to explain the high NH$_3$ loading over the IGP.

Apart from dry deposition and wet removal by precipitation, another main sink for NH$_3$ is scavenging by acidic species to form particulate NH$_4^+$. H$_2$SO$_4$ and HNO$_3$ resulting from the oxidation of sulfur dioxide (SO$_2$) and nitrogen oxides (NO$_x$) are major acidic species in the atmosphere. Previous studies have confirmed that reduced SO$_2$ and NO$_x$ emissions are key factors driving the increase in NH$_3$ concentration (Liu et al., 2018;Yu et al., 2018;Warner et al., 2017). In addition, meteorological conditions (including wind speed, precipitation, relative humidity and air temperature) also influence NH$_3$ loading through various chemical and physical processes. These factors may be causing the high NH$_3$ loading over the IGP, but these assumptions have not been verified in a modeling study.

In this study, we use a regional air quality model to investigate the causes of high NH$_3$ loading over the IGP during pre-monsoon and monsoon seasons. This is the first study to analyze the causes of high NH$_3$ loading over the IGP. The remainder of this paper is organized as follows. The air quality model and observational data are described in sect. 2. Section 3 analyzes the influences of several factors (including emissions, chemical conversion, and meteorological conditions) on NH$_3$ loading. Among them, SO$_2$ and NO$_x$ emissions over the IGP are compared to those over the NCP to clearly illustrate their impacts on NH$_3$ loading. Section 4 provides concluding remarks.

## 2 Methods

### 2.1 WRF-Chem model and emissions inventory

WRF-Chem (Fast et al., 2006;Grell et al., 2005) version 3.6.1 was applied to investigate the cause of the high NH$_3$ loading over the IGP during pre-monsoon and monsoon seasons. The simulation was performed on a domain with 30 km horizontal resolution covering the northern part of India and parts of Pakistan, Nepal, China, and Bangladesh with $120 \times 90$ grid cells. There were 23 vertical levels from the surface to the top pressure of 50 hPa. The simulations were conducted from 25 May to 31 August 2010 and the first 7 days (25-31 May) were treated as the spin-up period. June was considered pre-monsoon season. July to August was considered monsoon season. The initial meteorological and boundary conditions were obtained from the National Centers for Environmental Prediction Final Analysis with a 6 h temporal resolution. CBM-Z (Carbon Bond Mechanism version Z) chemical mechanism (Zaveri and Peters, 1999) and MOSAIC (Model for Simulating Aerosol Interactions and Chemistry) aerosol module (Zaveri et al., 2008) were used for modeling gas phase photochemistry and aerosol processes (including nucleation, coagulation, condensation and thermodynamic equilibrium), respectively. Dry deposition for trace gases and aerosols was treated following the methods of Wesely (1989) and Binkowski and Shankar (1995), respectively. Wet deposition in the model includes both in-cloud and below-cloud scavenging. The below-cloud scavenging of aerosols and trace gases was calculated based on the methods of Easter et al. (2004). More model configurations are described in Table S1.

Anthropogenic emissions were obtained from the MIX inventory (Li et al., 2017), an Asian anthropogenic emissions inventory that harmonizes several local inventories using a mosaic approach. MIX uses Regional Emissions Inventory in Asia (REAS2, version 2) (Kurokawa et al., 2013) for $NH_3$ emissions in India.

## 2.2 Observational dataset

Atmospheric total columns of $NH_3$ were derived from measurements of an Infrared Atmospheric Sounding Interferometer
(IASI) on board MetOp-A (https://iasi.aeris-data.fr/NH3/). Metop-A was launched in 2006 in a Sun-synchronous orbit with a mean local solar overpass time of 9:30 a.m. and 9:30 p.m. Only the daytime measurements have been used here, because the nighttime measurements had larger relative errors caused by the general lower thermal contrast for the nighttime overpass (Van Damme et al., 2014). It has been found that the IASI samples at the overpass time could represent the entire day, and IASI $NH_3$ observations are in fair agreement with the available ground-based and airborne data sets around the world
(Dammers et al., 2016;Van Damme et al., 2015a). However, due to the lack of publicly available ammonia observation data sets in the IGP, previous studies have not evaluated IASI $NH_3$ in the IGP. This work used the ANNI-NH3-v2.2R-I retrieval product, which relied on ERA-Interim reanalysis for its meteorological inputs (Van Damme et al., 2017). The mean $NH_3$ column concentrations over East Asia on a $0.25° \times 0.25°$ grid from June to August 2010 have been determined based on the relative error weighting mean method (Van Damme et al., 2014). $SO_2$ columns from June to August 2010 were derived from
the Level-3 Aura/OMI Global $SO_2$ Data Products (OMSO2e) (Krotkov et al., 2015). Tropospheric $NO_2$ columns from Ozone Monitoring Instrument (OMI) aboard NASA Aura satellite were used from June to August 2010 (http://www.temis.nl/airpollution/no2col/no2regioomimonth_qa.php).

Meteorological data at 38 sites over northern India obtained from the National Climate Data Center (https://gis.ncdc.noaa.gov/maps/ncei/cdo/hourly) were used to evaluate the accuracy of meteorological simulations. The
evaluated variables included hourly wind speed at 10 m (WS10), wind direction at 10 m (WD10), relative humidity at 2 m (RH2) and temperature at 2 m (T2). The statistical parameters included the mean bias (MB), normalized mean bias (NMB), root mean square error (RMSE) and correlation coefficient (R). In addition, air temperature at 21 sites over the NCP obtained from the National Climate Data Center were also used in this work.

## 2.3 ISORROPIA-II thermodynamic model

The thermodynamic equilibrium model, ISORROPIA-II (Fountoukis and Nenes, 2007), treating the thermodynamics of $NH_4^+$-$SO_4^{2-}$-$NO_3^-$-$K^+$-$Ca^{2+}$-$Mg^{2+}$-$Na^+$-$Cl^-$-$H_2O$ aerosol system, was used to investigate the influence of air temperature on the $NH_3$ total columns. In this study, ISORROPIA-II was run in the "forward mode" and assuming particles are "metastable" with no solid precipitates. The chemical and meteorological data from WRF-Chem, including water-soluble ions ($SO_4^{2-}$, $NO_3^-$, $NH_4^+$, $Cl^-$, $Na^+$) in $PM_{2.5}$, gaseous precursors ($NH_3$, $HNO_3$, $HCl$), temperature (T) and relative humidity (RH) are used
as the inputs of ISORROPIA-II. Using ISORROPIA-II, we simulated 20 scenarios. In these cases, air temperature of each layer increased or decreased by 10 °C synchronously, with the interval of 1 °C. Meanwhile, the other input parameters

remained the same. Then, we calculated the columnar $\varepsilon(NH_4^+)$ (partitioning ratios of $NH_4^+$ to total ammonia (TA, TA = $NH_3$ + $NH_4^+$)) in each case. The columnar $\varepsilon(NH_4^+)$ is the sum of the $\varepsilon(NH_4^+)$ of each vertical level, but each weighted by the thickness of the layer and mass concentration of TA. Sensitivity tests were firstly conducted based on the average of the entire IGP from June to August. Then, the IGP was divided equally from northwest to southeast into three regions (namely western IGP, central IGP, and eastern IGP), and the study period was divided into the pre-monsoon season (June) and the monsoon season (July to August). Sensitivity tests were conducted for the three regions under the two seasons.

## 3 Results

### 3.1 High NH₃ emissions

IGP is a vast stretch of fertile alluvial plain spanning the banks of the Indus and Ganges Rivers and their tributaries. The main part of the IGP is located in India. The estimated $NH_3$ emissions in India in 2010 were 9.9 Tg, which is comparable to that in China (9.8 Tg) and accounts for about 34 % of total $NH_3$ emissions in Asia (Li et al., 2017). Agriculture is the largest $NH_3$ emitter in India, accounting for about 76 % of the total $NH_3$ emissions (Li et al., 2017). Agricultural $NH_3$ emissions mainly originate from animal husbandry and fertilizer application (Bouwman et al., 1997;Streets et al., 2003). India is the second largest N-fertilizer consumer (after China) and consumes 16.5 Tg N-fertilizers (16 % of the world's total) (FAOSTAT, 2010). In addition, there are an estimated 302 million cattle and buffalo in India (19 % of world's total), which is more than any other country (FAOSTAT, 2010). It is estimated that cattle and buffalo account for about 80 % of $NH_3$ emissions among livestock in India (Aneja et al., 2012).

$NH_3$ emissions over the IGP was 4.3 Tg in 2010 (estimated using the MIX database), which was mainly attributed to intensive agricultural practices. The IGP is known as the food bowl of India spreading across the states of Punjab, Haryana, Uttar Pradesh, Bihar, and West Bengal (blue quadrangle in Fig. 1). The total number of cattle and buffalo in the five states was estimated to be 103 million (34 % of the national total) in 2012 (GoI, 2012b). The total amount of N-fertilizer applied in the five states was estimated to be 6.9 Tg (42 % of the national total) in 2010 (GoI, 2012a). $NH_3$ emissions over the IGP from June to August are very high with a regional mean $NH_3$ emissions flux of 0.4 t $km^{-2}$ $month^{-1}$ (estimated using MIX database for 2010). This is consistent with satellite observations, which also show a peak of $NH_3$ columns over the IGP from June to August (Van Damme et al., 2015b). The peak of $NH_3$ emissions over the IGP might be the joint result of intensive N-fertilizer applications and high temperature. IGP has two cropping cycles including summer and winter (GoI, 2012a). June to August is one of the two main sowing periods in the IGP with a large amount of N-fertilizer applied to the cropland as base fertilizer. The monthly map of N-fertilizer application amounts from Nishina et al. (2017) shows that there are two peaks in N-fertilizer application amounts over the IGP with one in May-August, the other in November-December, which is consistent with the two cropping cycles in the IGP. In addition, the air temperature is very high over the IGP with an observed regional mean value of 30.9 °C from June to August 2010. Ammonia emissions increase exponentially with

temperature (Riddick et al., 2016). The high application rate of N-fertilizer and high air temperature could cause high $NH_3$ emissions, resulting in the high $NH_3$ columns.

The spatial distribution of mean $NH_3$ total columns over East Asia from June to August 2010 is shown in Fig. 1. The $NH_3$ columns over the IGP ($7.6 \times 10^{16}$ molecules cm$^{-2}$) were about twice as large as what was observed over the NCP ($4.1 \times 10^{16}$ molecules cm$^{-2}$). The NCP is also a large agricultural region (Huang et al., 2012). The regional mean $NH_3$ emissions flux over the NCP was 0.7 t km$^{-2}$ month$^{-1}$ from June to August 2010 (estimated using the MIX database), which was about 1.8 times that of the IGP. The IGP has much higher $NH_3$ total columns (Fig. 1) compared to the NCP, but lower $NH_3$ emissions

fluxes (Fig. S1a). Therefore, other factors might lead to the high $NH_3$ loading over the IGP besides high $NH_3$ emissions.

## 3.2 Low gas-to-particle conversion of $NH_3$

The emissions fluxes of $SO_2$ and $NO_x$ (both are 0.3 t km$^{-2}$ month$^{-1}$) over the IGP are only about one-fourth of that over the NCP (1.1 and 1.3 t km$^{-2}$ month$^{-1}$) (Table 1 and Fig. S1). Besides, the satellite-derived $SO_2$ and $NO_2$ columns over the IGP (0.5 and $2.3 \times 10^{15}$ molecules cm$^{-2}$) are also much lower than that over the NCP (10.4 and $8.3 \times 10^{15}$ molecules cm$^{-2}$) (Fig.

S2). The relatively low $SO_2$ and $NO_x$ emissions could be an important factor causing the high $NH_3$ columns over the IGP. In this study, we used the molar ratio ($R_{emis}$) of $NH_3$ emissions fluxes ($E_A$) to the sum of twice the $SO_2$ emissions fluxes ($E_S$) and $NO_x$ emissions fluxes ($E_N$) to roughly represent the excess of $NH_3$ in the atmosphere, given by Eq. (1):

$$R_{emis} = \frac{E_A}{2 \times E_S + E_N} \; .$$

(1)

The calculated $R_{emis}$ in the IGP was 1.35, which was about 2.6 times as large as that in the NCP (0.51). We performed

simulations for a base case and a 'increased $SO_2$/$NO_x$ emissions' case to investigate the impact of $SO_2$ and $NO_x$ emissions on $NH_3$ loading. In the increased $SO_2$/$NO_x$ emissions case, the emissions of $SO_2$ and $NO_x$ increased 2.6 times to make $R_{emis}$ of the IGP equal to that of the NCP.

The simulated $NH_3$ columns in the base case are shown in Fig. 2a. It is noted that the IASI $NH_3$ columns cannot be quantitatively compared to modeled $NH_3$ columns as the IASI $NH_3$ products do not provide information on the vertical

sensitivity (averaging kernels) to properly weight the model values. Nonetheless, the simulated regional mean $NH_3$ total column over the IGP from the base case of $8.8 \times 10^{16}$ molecules cm$^{-2}$ is close to the satellite-derived value ($7.6 \times 10^{16}$ molecules cm$^{-2}$), indicating that the model could generally capture the magnitude of $NH_3$ columns. Additionally, a broadly similar pattern was found in the $NH_3$ columns in the base run as in the satellite observations, both of which showed that $NH_3$ columns decrease along the IGP from northwest to southeast with the highest values in the northwestern IGP (Figs. 1 and 2a).

The statistical performance of the meteorological predictions at 38 sites over Northern India are presented in Table S2. The predicted T2 matched well with the observations with a correlation coefficient of 0.8 and an NMB of 4.2 %. The predicted RH2 was slightly underestimated with an NMB of –13.4 % and a correlation coefficient of 0.8. The predicted WS10 agreed reasonably well with the observations with an NMB of –5.3 %. In addition, the simulated WD10 matched well with the

observations, and both the predicted and observed dominant wind direction was SSE. The good agreement between the simulation and the observations confirms the reliability of the meteorological prediction over the simulation domain.

The spatial distribution of the $NH_3$ total column in the increased emissions case is shown in Fig. 2b. The $NH_3$ total columns significantly decreased over the entire IGP, with a regional mean value of $2.5 \times 10^{16}$ molecules cm$^{-2}$ (a 72.2 % decrease compared to the base case). The surface $\varepsilon(NH_4^+)$ in the base case and the increased $SO_2/NO_x$ emissions case are shown in Fig. 2 (panels c and d, respectively). The surface $\varepsilon(NH_4^+)$ in the base case was low with a regional mean value of 0.3 over the IGP. The simulated $\varepsilon(NH_4^+)$ in the 2010 monsoon in Delhi was 0.38, which is close to the observed $\varepsilon(NH_4^+)$ (0.39 in the 2011 monsoon season in Delhi) (Singh and Kulshrestha, 2012). In the increased $SO_2/NO_x$ emissions case, the regional mean surface $\varepsilon(NH_4^+)$ increased to 0.6 over the IGP. Significant increases in surface $SO_4^{2-}$ and $NO_3^-$ concentrations were also found (Fig. S3). The regional mean surface $SO_4^{2-}$ and $NO_3^-$ concentrations increased from 9.7 to 24.9 and from 7.2 to 20.0 µg m$^{-3}$, respectively. Additionally, the regional mean columnar $\varepsilon(NH_4^+)$ over the IGP is 0.56 in the base case and increases to 0.87 in the increased $SO_2/NO_x$ emissions case. This suggests that the increased $SO_2$ and $NO_x$ emissions enhanced the formation of acidic species and promoted the conversion of $NH_3$ into $NH_4^+$. The effectively reduced $NH_3$ total columns in the increased $SO_2/NO_x$ emissions case indicate that low $SO_2$ and $NO_x$ emissions could be the major cause of the high $NH_3$ loading over the IGP.

Besides the amount of acidic species, air temperature is also an important factor affecting the thermodynamic equilibrium of $NH_3$ between the gas phase and the particle phase. Higher air temperature limits the gas-to-particle conversion of $NH_3$ and enhances volatilization of $NH_4NO_3$ (Seinfeld and Pandis, 2006). The observed regional mean air temperature over the IGP from June to August 2010 was 30.9 °C, about 4.9 °C higher than the NCP (26.0 °C). As shown in Fig. 3a, the columnar $\varepsilon(NH_4^+)$ increases as temperature decreases. A 10°C decrease in temperature results in a 0.07 increase in $\varepsilon(NH_4^+)$ and a consequent 17 % decrease in $NH_3$ total columns. Additionally, a 10°C increase in temperature results in a 0.08 decrease in $\varepsilon(NH_4^+)$ and a consequent 20 % increase in $NH_3$ total columns. If the temperature over the IGP drops to the temperature typical of the NCP (a 4.9 °C decrease), the $NH_3$ total columns over the IGP will only decrease by 10 %. In contrast, if the $SO_2/NO_x$ emissions over the IGP increase to make the $R_{emis}$ of the IGP equal to that of the NCP, the $NH_3$ column over the IGP will decrease by 72.2 %. Therefore, the low $SO_2/NO_x$ emissions have a greater effect on causing high $NH_3$ columns over the IGP than the high air temperature. As shown in Figure 3c, the sensitivity of $NH_3$ to temperature varies in different seasons and regions. Temporally, the sensitivity of $NH_3$ to temperature during the monsoon season is generally higher than that during the pre-monsoon season. Spatially, the sensitivity of $NH_3$ to temperature is highest over the eastern IGP, followed by the central IGP and the western IGP. The difference in the sensitivity of the $NH_3$ to temperature may be caused by the difference of the initial $\varepsilon(NH_4^+)$ and temperature.

### 3.3 Weak horizontal diffusion of $NH_3$

The IGP is surrounded by unique topography with the Himalayan range to the north and the Sulaiman range to the west. Weather on the Indian subcontinent is controlled by the low-level Indian monsoon regime from June through September

(Lawrence and Lelieveld, 2010). Fig. 4a shows the spatial distributions of surface wind flow and wind speed from June to August 2010. The dominant wind direction is southwest over the Indian peninsula and southeast over the IGP. Air mainly flows from the west coast of India and the south coast of Bengal. Surface wind speed is high on the west coast of India (>5 m s$^{-1}$) and on the south coast of Bengal (>4 m s$^{-1}$) but decreases from the coast inland. Mountains serve as barriers to the airflow on the surface of the Earth (Barry, 2008). Chow et al. (2013) reported that when stably stratified airflow encounters an extra-tropical mountain barrier, it is forced to rise and cool adiabatically. Consequently, higher pressure along the slope could be created, which could decelerate and block the flow. After a while, geostrophic adjustment occurs. As a result, the airflow turns left (right) in the northern (southern) hemisphere, and a barrier jet blowing parallel to the barrier is formed. As shown in Fig. 4a, the southerly airflow from the Bay of Bengal turns left when approaching the Himalayas, and then an easterly barrier jet parallel to the Himalayas is formed. The southwesterly airflow from the west coast of India also turns left when approaching the Himalayas. Similarly, wind flow at 850 hPa (Fig. 4b) also shows left-turning airflow near the Himalayas. The left-turning airflow indicates that the barrier effect of the Himalayas limits the northward movement of polluted air. Both satellite observations and the model simulation show that the high $NH_3$ columns over the IGP are effectively cut off by mountains to the north (Fig. 2a and Fig. S4).

As shown in Fig. 4b, an area of low geopotential height extends from Pakistan to east India following the IGP. This elongated region of low pressure is known as the monsoon trough (Bohlinger et al., 2017). It causes wind to converge over this region. The convergence of horizontal wind can be observed from wind flow at both the surface and at 850 hPa. The prevailing wind directions south and east of the IGP are southwest and southeast, respectively. As a result of convergence of horizontal winds, an area of low wind speed forms and covers most of the IGP. The regional mean surface wind speed over the IGP is <3 m s$^{-1}$. The weak wind speed in association with the convergence weakens the horizontal advection of $NH_3$ and results in the accumulation of $NH_3$ over the IGP.

The ventilation rate ($V_r$) of the four edges of the IGP was used to illustrate the accumulation of an air mass over the IGP (Fig. 4d). The $V_r$ of one edge is defined as the product of sectional area to the transport wind, given by Eq. (2):

$$V_r = AU_T .$$
(2)

The sectional area A can be expressed as A = ZL, where Z is the mean planetary boundary layer (PBL) height along the edge and L is the length of the edge. The transport wind $U_T$ is given by $U_T = \frac{1}{m}\sum_{j=1}^{m}\left(\frac{1}{n}\sum_{i=1}^{n}U_{ij}\right)$ . m and n are the number of locations along the edge and vertical levels within the PBL where the winds are measured or predicted. $U_{ij}$ is the wind speed perpendicular to the cross-section at each height and location along the edge. The ventilation rates of the four edges of the IGP were calculated using the WRF-Chem simulation results. The total $V_r$ of the inflow from the southern and eastern edges ($3.1 \times 10^9$ m$^3$ s$^{-1}$) was 64 % higher than the total $V_r$ of the outflow from the western and northern edges ($1.9 \times 10^9$ m$^3$ s$^{-1}$).

The strong inflow and weak outflow indicate accumulation of the air mass over the IGP. Therefore, outward transport of $NH_3$ from the IGP through horizontal advection could be weak.

Interestingly, both relative humidity and precipitation are high over the IGP (Figs. S5), with regional mean values of 63 % and 660 mm from June to August 2010. The high relative humidity and precipitation suggest strong gas-to-particle conversion and wet scavenging of $NH_3$ (Seinfeld & Pandis, 2006). The observed high $NH_3$ loading under such a wet condition further indicates the effectiveness of other factors leading to high $NH_3$ loading. The simulated surface $\varepsilon(NH_4^+)$ over the western, central and eastern part of the IGP were 0.11, 0.13 and 0.24 during pre-monsoon and 0.26, 0.26 0.37 during monsoon. It is not difficult to find that the surface $\varepsilon(NH_4^+)$ during the monsoon season is significantly higher than that during the pre-monsoon season, and the surface $\varepsilon(NH_4^+)$ generally increases from northwest to southeast along the IGP. Besides, the columnar $\varepsilon(NH_4^+)$ shows similar spatiotemporal variations with the surface $\varepsilon(NH_4^+)$ (Figure 3b). The spatiotemporal variations of $\varepsilon(NH_4^+)$ are consistent with the spatiotemporal variations of RH (Figure S5a), indicating that RH is an important factor affecting the $NH_3$ partitioning. The meteorological conditions in the northwest IGP are characterized by higher air temperature, lower humidity, and lower rainfall compared to the southeast IGP (Figs. 4c and S5), all of which are conducive to the increase of $NH_3$. Consistently, $NH_3$ total columns decrease from northwest to southeast along the IGP as revealed by both the satellite measurements and model simulations (Figs. 1 and 2a). However, emission fluxes of $NH_3$ over the northwest IGP are also obviously higher than the southeast IGP (Fig. S1). To exclude the impact of emissions on the spatial distributions of $NH_3$, simulations for a "homogeneous emissions" case was performed by using WRF-Chem, where emissions of all primary pollutants over the IGP were set to their regional mean values. As shown in Fig. 5, $NH_3$ total columns in the homogeneous emissions case still appear to decrease from northwest to southeast along the IGP. It is indicated that the meteorological factors (atmospheric diffusion, temperature, relative humidity, and precipitation) are important causes of the higher $NH_3$ loadings over the northwest IGP than the southeast IGP.

## 4 Conclusions

Satellite observations have revealed that the IGP has the global maximum $NH_3$ loading with a peak from June to August. Our study reveals that the high $NH_3$ loading over the IGP appears to be the joint result of high $NH_3$ emissions, weak chemical loss, and weak horizontal diffusion. Intensive agricultural activities in combination with high temperature resulted in relatively high $NH_3$ emissions over the IGP, with a regional mean $NH_3$ emissions flux of 0.4 t $km^{-2}$ $month^{-1}$. The low $SO_2$ and $NO_x$ emissions and high temperature limited the conversion of $NH_3$ to $NH_4^+$, which is a key reason for the high $NH_3$ loading over the IGP. In addition, orographic and meteorological conditions also play important roles in $NH_3$ accumulation over the IGP. The barrier effects of the Himalayas limit the northward movement of monsoon air. The low wind speed (<3 m $s^{-1}$) in association with the surface convergence over the IGP weakens horizontal diffusion, which is conducive to the accumulation of $NH_3$ over the IGP.

The gas-particle partitioning plays an important role in influencing $NH_3$ columns. The deviation of the simulated sulfate and nitrate will cause a deviation in the simulated $NH_3$ by affecting $NH_3$ gas-particle partitioning. Thus, in addition to the $NH_3$ and $NH_4^+$, the simulated concentrations of sulfate and nitrate are also necessary to be constrained using field observations in the future. Besides, organic species are not considered in the thermodynamic calculations in this study, because the impact of organic species on aerosol thermodynamics is still rather poorly understood (Zaveri et al., 2008;Fountoukis and Nenes, 2007)). Pye et al. (2018) found that the AIOMFAC (Aerosol Inorganic–Organic Mixtures Functional groups Activity Coefficients) based equilibrium model considering inorganic-organic interactions was consistent with ISORROPIA in terms of $NH_3$ gas-particle partitioning. Metzger et al. (2006) found that the $\varepsilon(NH_4^+)$ calculated by ISORROPIA was about 15% lower than that calculated by EQSAM2 (Equilibrium Simplified Aerosol Model) considering organic acids. Thus, the influence of organic species on the $NH_3$ gas-particle partitioning might be limited and will not have a significant impact on the results of this study. However, these two studies were conducted in the United States. The effects of organic species on aerosol thermodynamics in the IGP need further research in the future. Additionally, dry and wet deposition also has an important influence on $NH_3$ columns. Field observations of the dry and wet deposition of $NH_3$ and $NH_4^+$ in the IGP are needed to constrain model simulations in the future.

**Data availability**

The IASI data used in this study was provided by the AERIS data infrastructure (https://iasi.aeris-data.fr/NH3/). The meteorological data used in this study was obtained from the National Climate Data Center integrated surface database (https://gis.ncdc.noaa.gov/maps/ncei/cdo/hourly). The anthropogenic emissions are available from MIX inventory (www.meicmodel.org/dataset-mix). The $SO_2$ columns were provided by the NASA Goddard Earth Sciences Data and Information Services Center (https://disc.gsfc.nasa.gov/datasets/OMSO2e_003/summary). The $NO_2$ columns are available from the Tropospheric Emission Monitoring Internet Service (http://www.temis.nl/airpollution/no2col/no2regioomimonth_qa.php).

**Author contribution**

Y.S initiated the investigation. T.W performed the modelling analyses. T.W, Y.S, Z.X and T.Z wrote and edited the manuscript. M.L, T.X, W.L, L.Y, X.C, H.Z and L.K contributed to discussions of the results and the manuscript.

**Competing interests**

The authors declare no competing interests.

## Acknowledgements

This work was supported by the National Natural Science Foundation of China (NSFC) (91644212, 41675142 and 91837209) and the National Key R&D Program of China (2016YFC0201505).

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

**Table 1.** Regional Mean $NH_3$ Total Columns and Emissions Fluxes of $NH_3$, $SO_2$, and $NO_x$ of the IGP and the NCP from June to August 2010.

| | $NH_3$ total columns[a] (molecules $cm^{-2}$) | Emissions fluxes[b] (t $km^{-2}$ $month^{-1}$) | | |
| --- | --- | --- | --- | --- |
| | | $NH_3$ | $SO_2$ | $NO_x$ |
| IGP | $7.6 \times 10^{16}$ | 0.4 | 0.3 | 0.3 |
| NCP | $4.1 \times 10^{16}$ | 0.7 | 1.1 | 1.3 |

[a]$NH_3$ total columns were derived from IASI measurements

[b]Emissions fluxes were estimated using the MIX database

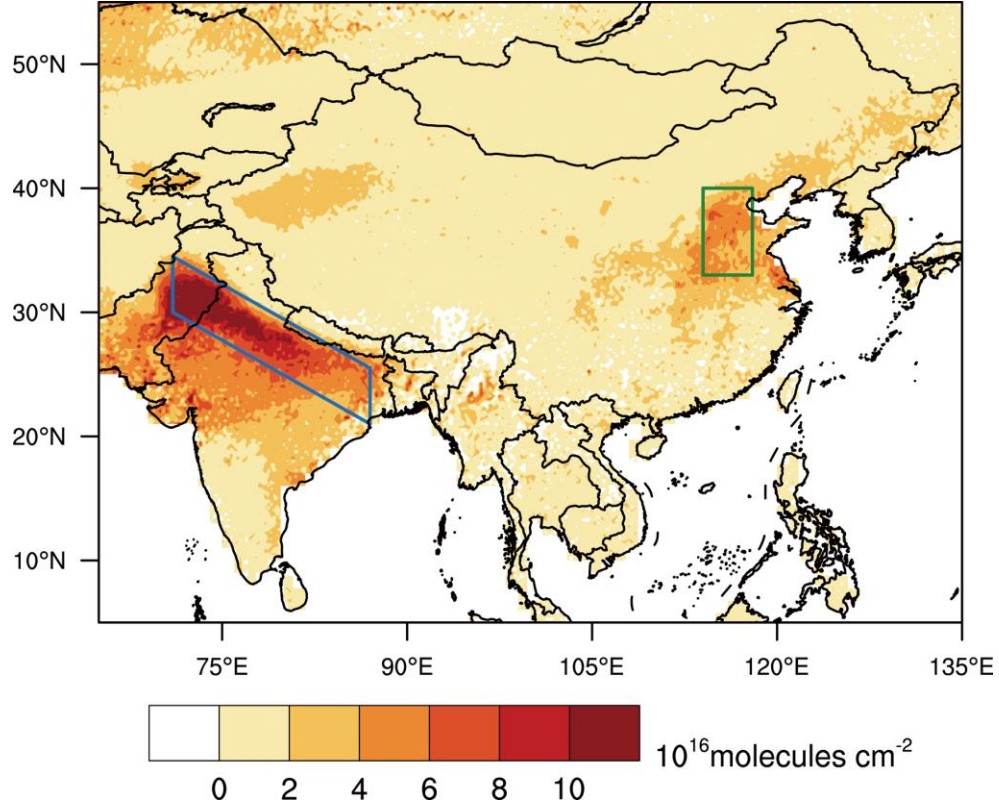

**Figure 1.** The spatial distribution of $NH_3$ total columns over East Asia from June to August 2010 retrieved from IASI measurements. The blue quadrangle represents the Indo-Gangetic Plain (IGP), and the green quadrangle represents the Northern China Plain (NCP).

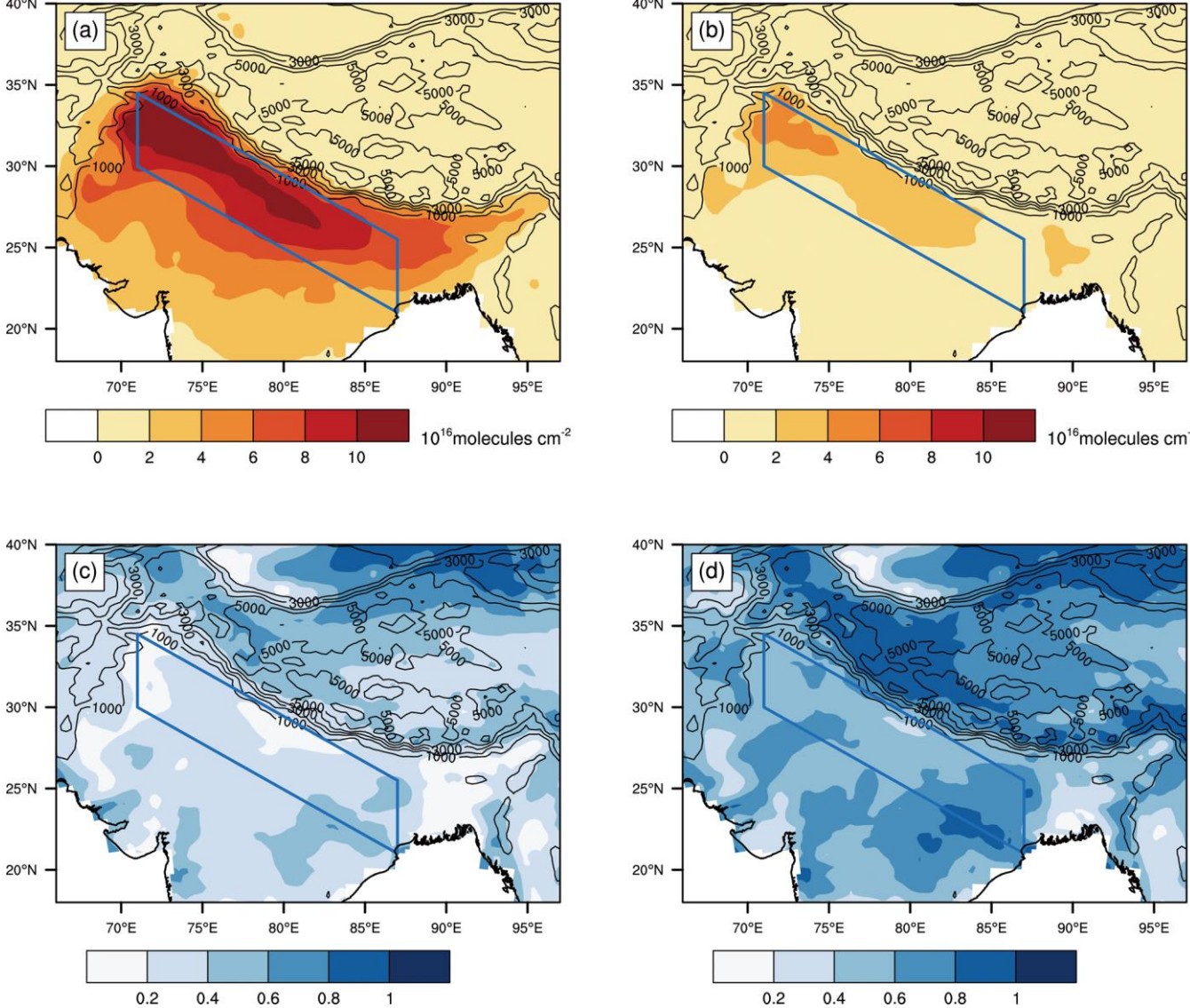

**Figure 2.** Spatial distributions of WRF-Chem predicted total columns of NH₃ and surface ε(NH₄⁺) from June to August 2010. (a) and (b) are total columns of NH₃ in the base case and the increased SO₂/NOₓ emissions case, respectively. (c) and (d) are surface ε(NH₄⁺) in the base case and the increased SO₂/NOₓ emissions case, respectively.

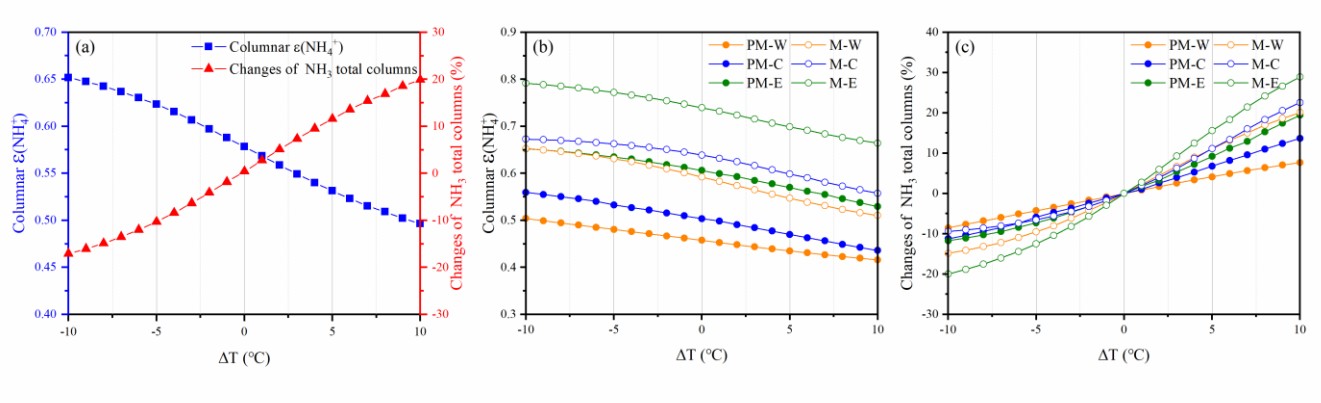

**Figure 3.** Columnar ε(NH₄⁺) and changes of NH₃ total columns with the changes of temperatures predicted by ISORROPIA-II. (a) Mean columnar ε(NH₄⁺) and changes of NH₃ total columns over the IGP from June to August 2010. (b) Columnar ε(NH₄⁺) and (c) changes of NH₃ total columns over the western IGP during Pre-monsoon (PM-W), the central IGP during Pre-monsoon (PM-C), the eastern IGP during pre-monsoon (PM-E), the western IGP during monsoon (M-W), the central IGP during monsoon (M-C), the eastern IGP during monsoon (M-E).

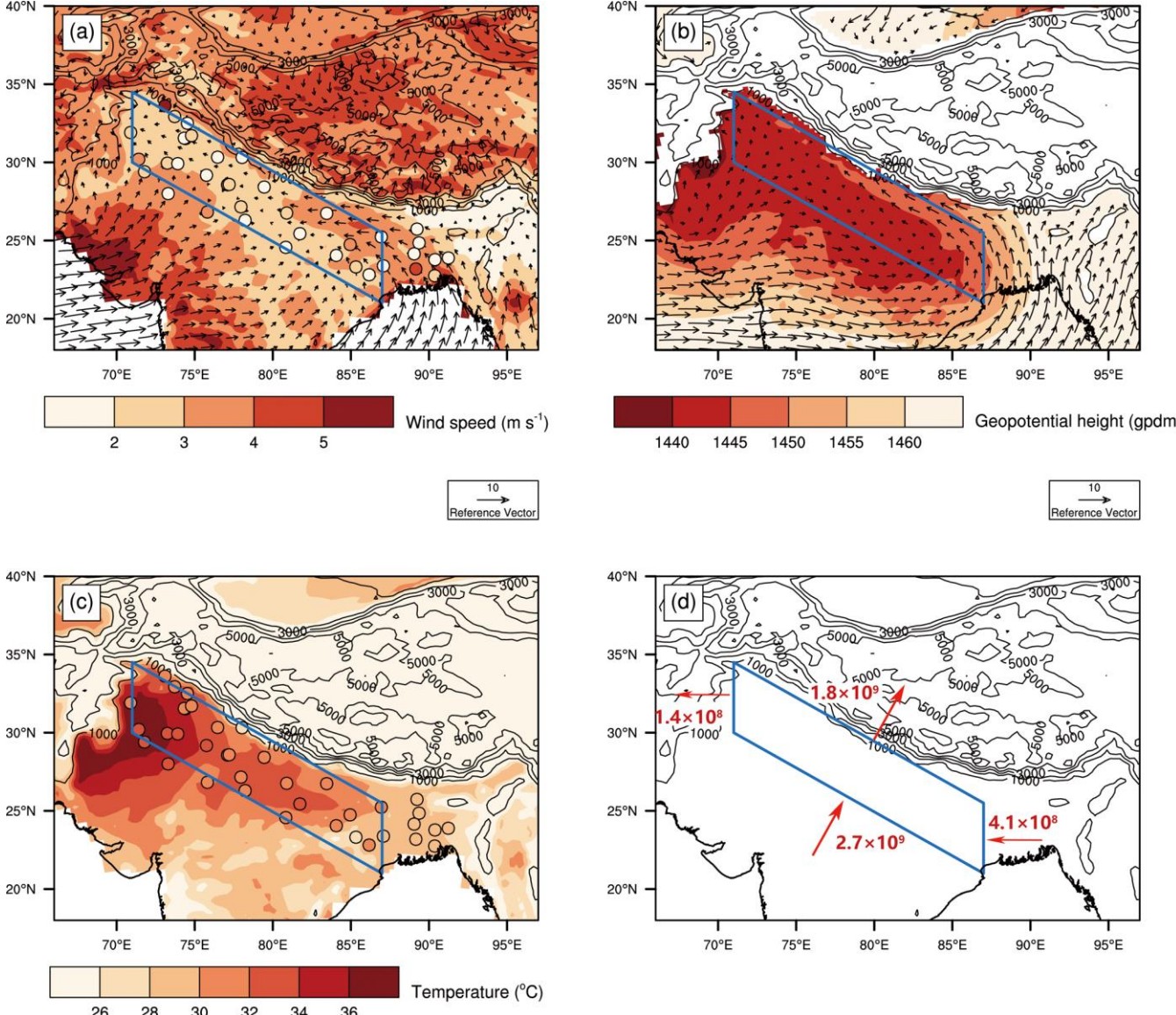

**Figure 4.** Spatial distributions of WRF-Chem predicted meteorological variables from June to August 2010. (a) Wind flow and wind speed at 10 m. (b) Wind flow and geopotential height at 850 hPa. (c) Air temperature at 2 m. (d) Ventilation rate ($m^3$ $s^{-1}$) of the four edges of the IGP. Circles in (a) and (c) show the observed wind speed at 10 m and air temperature at 2 m, respectively.

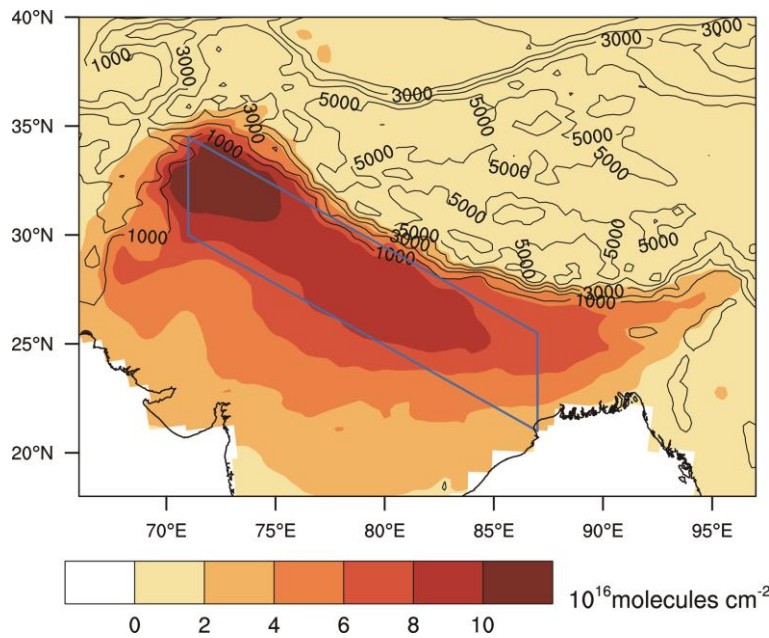

**Figure 5.** Spatial distributions of WRF-Chem predicted total columns of NH$_3$ from June to August 2010 in the homogeneous emissions case.