# Peer review of "Why the Indo-Gangetic Plain is the region with the largest NH3 column in the globe during pre-monsoon and monsoon seasons?"

_Atmospheric Chemistry and Physics, 2019_

## Referee Comment (RC1) · Anonymous Referee #1 · 18 Jan 2020

**Review of "Why the Indo-Gangetic Plain is the region with the largest NH$_3$ column in the globe during summertime?"**

**General Comments**

This study aims to explore the reasons behind the elevated levels of ammonia observed over the Indo-Gangetic Plain. This is an important and scientifically relevant question, particularly since the ammonia burden has significant implications on inorganic aerosol concentrations over the region. The authors use the WRF-Chem model to investigate the physics and thermodynamics underlying the atmospheric fate of NH$_3$ and the resulting analysis provides useful insights into some of the factors driving the high concentrations over the region. For instance, the demonstrated sensitivity to increasing NOx and SO$_2$ emissions is an interesting result, particularly when contrasted to the case over the North China Plain.

While the dynamic physical transport and meteorology simulated by the model is validated and constrained, the approach chosen for the thermodynamic analysis largely hinges on an offline test of emissions and temperature sensitivities using the ISORROPIA model. The analysis would benefit significantly from a more rigorous on-line treatment of the thermodynamics that investigates the specific factors controlling NH$_3$ partitioning in greater detail. In addition, observational constraints (such as satellite ammonia columns) could be further leveraged to get a more quantitative estimate of model performance prior to interpreting the model output. With this in mind, I provide the following comments below and recommend that these issues be addressed prior to publication in ACP.

**Specific Comments**

**Model Details, Validation and Uncertainties**

In order to appropriately interpret the WRF-Chem analysis the authors should provide a more detailed discussion on the specifics of the ammonia simulation in WRF-Chem. For instance, is it possible to run ISORROPIA in an online configuration (partitioning at each time-step and explicitly simulating the aerosol species)? Unless I am misinterpreting the methods section, it appears that ISORROPIA is used only in an offline context. Given the spatial and temporal heterogeneity in the various factors that drive aerosol partitioning, running ISORROPIA in an online configuration would more appropriately explore the scientific questions outlined in this study.

*Line 70:* The authors have validated the simulated meteorology (wind speed, temperature, etc.) but do not validate the ammonia simulation itself. While I recognize this is challenging, it could potentially be done using satellite measurements (with the appropriate application of an averaging kernel) or surface measurements where available. Even a general estimate of how well the model captures ammonia variability and magnitude over the region would provide important context. Ideally, the different (non-transport) factors that dictate ammonia concentrations (namely – emissions, wet deposition, dry deposition and aerosol portioning) would be constrained using observational data whenever available. In absence of such data, an explanation of the uncertainties associated with these various processes (and the steps that need to be taken to constrain them) is required in order to appropriately interpret the results.

*Line 53:* The simulations described here are spun-up over a 7-day period. A more detailed discussion about the initial concentrations assumed for the most important chemical species and their estimated

lifetimes would provide useful context on whether the week-long period provides sufficient time to allow the longer-lived gas-phase species to equilibrate prior to the main simulation period.

**Seasonal Analysis**

*Line 52:* The authors classify June to August as the summer period. However, in India, this season is characterized by the monsoons (usually beginning in mid-June) which are associated with drastic changes to regional meteorology. This perhaps provides context for the statement on Line 198, given that high levels of precipitation and humidity are expected during the monsoon season. The Indian summer is usually thought to be between the months of April – May.

**Impact of Transport and Meteorology**

Section 3.2 discusses the importance of various meteorological drivers (such as RH and temperature). However, given its importance in determining ammonia burdens, a more detailed discussion of the specific mechanisms that dictate aerosol partitioning under different meteorological conditions and the associated uncertainties in our understanding of these processes would add to the broader utility of this study.

**Use of the ISORROPIA module to access the impact of emissions and temperature**

*Line 77:* A more detailed overview of the ISORROPIA module would provide important context for the resulting analysis. If only applied in an offline context, it is possible that the analysis is not capturing various important (and non-linear) effects due to the spatial heterogeneity in gas phase and particle concentrations (along wih the associated depositional losses at every timestep). While the authors provide an observational constraint (Line 142) to validate this approach, the differences in the model and observed partitioning ratio are significant (on the order of 30%). A more thorough comparison with the observational data would greatly benefit the analysis and serve as validation for some of the later conclusions. Additionally, the comparison of the regional mean to observational data over Delhi may not be appropriate, particularly given that NOx and $SO_2$ concentrations are likely much higher over the city.

*Line 155:* The temperature sensitivity is an interesting result, particularly when contrasted to the $SO_2$/NOx sensitivity. However, the approach here considers only a simplified case over the entire region. If the partitioning was conducted online (at every timestep), would it be reasonable to expect a different sensitivity to changes in temperature / $SO_2$ / NOx? A more detailed discussion about the non-linear, spatially dependent factors driving aerosol concentrations (and the heterogeneity in emissions, loss processes, thermodynamics, etc.) would provide more context with which to interpret these results.

**A discussion of other drivers of aerosol formation (particularly in the context of the monsoons)**

Singh and Kulshrestha (2012), cited in this study, hypothesized that humidity during the monsoon season had a significant impact on $NH_3$ partitioning. Could the authors discuss this in the context of their analysis? Given that the aim of this study is to establish the most salient drivers of the high $NH_3$ concentrations (particularly during the selected monsoon period), a more detailed discussion of what determines the relative dominance of the different production, loss and partitioning mechanisms under various atmospheric conditions would provide important context with which to interpret the results of this analysis.

---

## Referee Comment (RC2) · Anonymous Referee #2 · 21 Jan 2020

General comments

The study analyses the possible factors responsible for elevated levels of ammonia over Indo-Gangetic Plain (IGP) regions during summertime. From previous studies, IGP has been identified as a hotspot for ammonia and analyzing the reasons for the high levels is an important study. Considering the implications of high-level ammonia, this study improves the understanding of the scientific community working on the ammonia over IGP. The authors use the air quality model (WRF-Chem), emission inventory and observation data (IASI satellite) to analyze the possible factors namely chemical conversion, emissions, and meteorology on ammonia loading.

Specific comments with line numbers are provided below and I recommend these issues to be addressed before publication in ACP.

Specific comments

Title: The authors classify June to August as the summer period in the study. However, the Indian summertime is considered from April-June and monsoon begins from mid-June/late June. Summer to monsoon season has a drastic change in regional meteorology over IGP and possibility impact the levels of ammonia (gas phase) over IGP from high to low because of washout effects? I would suggest the authors not to mention the summer season in the title or in the study in general if possible or provide some explanation here.

Methods

Line 65: The authors mentioned the fair agreement of IASI ammonia observations with ground-based measurements citing few studies which are mainly satellite observations. It is not clear whether the fair agreement was for IGP or other regions. It would be helpful if the authors could provide some details about the ground measurements used here.

Line 77: The authors should provide more details on the input of the ISOROPIA-II model or in general the model in order to interpret the gas-particle partitioning of the ammonia for example what are the gas species in input?

Line 80: "As inputs of ISORROPIA-II, the outputs (water-soluble ions, gas species, T and RH) of WRF-Chem were first averaged over the IGP and then averaged for summer 2010". Please provide more details on averaging here. As IGP is a vast region, averaging the inputs over the region may create a bias for some regions over IGP considering the heterogeneity of the sources over IGP. Again the time averaging may create some bias too as the time period included in the study has few days or a month of the summer season and two months of monsoon as per Indian meteorological

department classification. It would be useful to check the variability of the inputs month wise for example June, July, and August separately.

Results

Line 105: As per the sowing season, IGP has mainly two cropping cycles which includes summer and winter both. During both cycles, fertilizer applications can be intensive. Is there any study/data supporting the highest N- fertilizer application during the months mentioned in this study? Line108: Authors cited the Riddick et al. 2010 for the exponential increase of ammonia emission with temperature. Would it be possible to expect similar results, if the temperature data and ammonia satellite observations from the present study taken in to account?

Line 160: The analysis of the low gas-to-particle conversion of ammonia demonstrates the sensitivity to SO2/NOx emissions. This is an excellent analysis based on the model output data. Is it fair to expect similar results if the data for SO2/NOx not modeled but taken from observations either ground or satellite observations? Please comment on this.

Line 200: The authors concluded an interesting observation about the other factors than wet conditions and high RH controlling the high ammonia loading over the IGP. More details on this would be useful to support this conclusion. Even a simple time series of IASI ammonia observations with RH for the period of study would provide an important context.

---

## Author Comment (AC1) · 15 Apr 2020

Our point-by-point responses are provided below. The referees' comments are italicized.

**Response to Referee #1**

*Referee: This study aims to explore the reasons behind the elevated levels of ammonia observed over the Indo-Gangetic Plain. This is an important and scientifically relevant question, particularly since the ammonia burden has significant implications on inorganic aerosol concentrations over the region. The authors use the WRF-Chem model to investigate the physics and thermodynamics underlying the atmospheric fate of NH3 and the resulting analysis provides useful insights into some of the factors driving the high concentrations over the region. For instance, the demonstrated sensitivity to increasing NOx and SO2 emissions is an interesting result, particularly when contrasted to the case over the North China Plain.*

*While the dynamic physical transport and meteorology simulated by the model is validated and constrained, the approach chosen for the thermodynamic analysis largely hinges on an offline test of emissions and temperature sensitivities using the ISORROPIA model. The analysis would benefit significantly from a more rigorous online treatment of the thermodynamics that investigates the specific factors controlling NH3 partitioning in greater detail. In addition, observational constraints (such as satellite ammonia columns) could be further leveraged to get a more quantitative estimate of model performance prior to interpreting the model output. With this in mind, I provide the following comments below and recommend that these issues be addressed prior to publication in ACP.*

**Response:** We would like to thank the referrer for your detailed and constructive comments. Please see our point-by-point reply below.

*Referee: 1. Model Details, Validation and Uncertainties*

*In order to appropriately interpret the WRF-Chem analysis the authors should provide a more detailed discussion on the specifics of the ammonia simulation in WRF-Chem. For instance, is it possible to run ISORROPIA in an online configuration (partitioning at each time-step and explicitly simulating the aerosol species)? Unless I am misinterpreting the methods section, it appears that ISORROPIA is used only in an offline context. Given the spatial and temporal heterogeneity in the various factors that drive aerosol partitioning, running ISORROPIA in an online configuration would more appropriately explore the scientific questions outlined in this study.*

**Response:** In fact, the MOSAIC (Model for Simulating Aerosol Interactions and Chemistry) aerosol module embedded in WRF-Chem is used for online calculation of aerosol partitioning. Since WRF-Chem is a coupled model, variables such as air temperature cannot be arbitrarily perturbed. Thus, ISORROPIA-II is used offline to study the effect of air temperature on the gas-particle partitioning of $NH_3$. As you suggested, we added more details about the WRF-Chem simulation in Sect 2.1.

**Revision:** (Page 2, Line 57-63) "CBM-Z (Carbon Bond Mechanism version Z) chemical mechanism (Zaveri and Peters, 1999) and MOSAIC (Model for Simulating Aerosol Interactions and Chemistry) aerosol module (Zaveri et al., 2008) were used for modeling photochemical reactions and aerosol processes (nucleation, growth and thermodynamic equilibrium), respectively. Wet deposition in the model includes both in-cloud and below-cloud scavenging. The blow-cloud scavenging of aerosols by impaction and trace gases by mass transfer was based on the methods of Easter et al. (2004). Dry deposition was treated following the methods of Wesely (Wesely, 1989) for trace gases, and Binkowski and Shankar (Binkowski and Shankar, 1995) for aerosols."

*Line 70: The authors have validated the simulated meteorology (wind speed, temperature, etc.) but do not validate the ammonia simulation itself. While I recognize this is challenging, it could potentially be done using satellite measurements (with the appropriate application of an averaging kernel) or surface measurements where available. Even a general estimate of how well the model captures ammonia variability and magnitude over the region would provide important context. Ideally, the different (non-transport) factors that dictate ammonia concentrations (namely – emissions, wet deposition, dry deposition and aerosol portioning) would be constrained using observational data whenever available. In absence of such data, an explanation of the uncertainties associated with these various processes (and the steps that need to be taken to constrain them) is required in order to appropriately interpret the results.*

**Response:** It is inappropriate to quantitatively compare IASI $NH_3$ columns with modeled $NH_3$ columns. Because the IASI-$NH_3$ retrieval does not produce an averaging kernel to properly weight the model values, and using a typical averaging kernel will cause more uncertainty as there is a large day-to-day variability in the averaging kernels (Clarisse et al., 2009). The published observation data of $NH_3$ ang $NH_4^+$ around 2010 over the IGP are almost all seasonal averages, and no monthly (or above) resolution data has been found. Since the simulated time range (June-August) is inconsistent with the time range of the seasonal average (April-June or March-June is summer; July-September or July-October is the monsoon season), the simulated value cannot be compared with the observation quantitatively. Nevertheless, as you suggested, we added a general evaluation of model simulation from $NH_3$ magnitude and spatial variations using IASI-$NH_3$ as a reference.

For the uncertainty brought by the dry and wet deposition, it is difficult to explore the uncertainty caused by this factor due to the lack of relevant observation data. However, the relevant researches have confirmed the relative accuracy of the existing dry and wet deposition parameterization in WRF-Chem (Easter et al., 2004;Wesely, 1989). In addition, for the impact of thermodynamic partitioning on ammonia concentrations, we have

studied the effect of temperature and humidity on aerosol partitioning which could further affect ammonia concentrations, see reply 5 and reply 7 for detail.

**Revision:** (Page 5, Line 146-152) "It is noted that the IASI $NH_3$ columns cannot be quantitatively compared to modeled $NH_3$ columns as the IASI $NH_3$ products do not provide information on the vertical sensitivity (averaging kernels) to properly weight the model values. Nonetheless, the simulated regional mean $NH_3$ total column over the IGP from the base case of $8.8 \times 10^{16}$ molecules $cm^{-2}$ is close to the satellite-derived value ($7.6 \times 10^{16}$ molecules $cm^{-2}$), indicating that the model could generally capture the magnitude of $NH_3$ columns. Additionally, a broadly similar pattern was found in the $NH_3$ columns in the base run as in the satellite observations, both of which showed that $NH_3$ columns decrease along the IGP from northwest to southeast with the highest values in northwestern IGP (Figs. 1 and 2a)"

*Line 53: The simulations described here are spun-up over a 7-day period. A more detailed discussion about the initial concentrations assumed for the most important chemical species and their estimated lifetimes would provide useful context on whether the week-long period provides sufficient time to allow the longer-lived gas-phase species to equilibrate prior to the main simulation period.*

**Response:** To explore the effect of spin-up period on the simulation results, we extended the spin-up period from one week to one month. Results show that extending the spin-up period caused $NH_3$ columns over the IGP to change by only 0.007%, indicating that the 7-day spin-up would not affect the $NH_3$ simulation.

*Referee: 2. Seasonal Analysis*

*Line 52: The authors classify June to August as the summer period. However, in India, this season is characterized by the monsoons (usually beginning in mid-June) which are associated with drastic changes to regional meteorology. This perhaps provides context for the statement on Line 198, given that high levels of precipitation and humidity are expected during the monsoon season. The Indian summer is usually thought to be between the months of April – May.*

**Response:** Accepted. As you suggested, we have corrected "summer" to "pre-monsoon and monsoon seasons". The title and the relevant part of main text have been revised.

*Referee: 3. Impact of Transport and Meteorology*

*Section 3.2 discusses the importance of various meteorological drivers (such as RH*

*and temperature). However, given its importance in determining ammonia burdens, a more detailed discussion of the specific mechanisms that dictate aerosol partitioning under different meteorological conditions and the associated uncertainties in our understanding of these processes would add to the broader utility of this study.*

**Response:** Accepted. To analyze the specific factors that affect $NH_3$ concentrations under different meteorological conditions, we divided the IGP from northwest to southeast into three regions (namely western IGP, central IGP, and eastern IGP), and divided the study period into two seasons (pre-monsoon and monsoon). Analyses of the sensitivity of $NH_3$ to temperature under different meteorological conditions have been added in Sect 3.2.

**Revision:** (Page 4, Line 98-101) "Sensitivity tests was firstly conducted based on the average of the entire IGP from June to August. Then, the IGP was divided equally from northwest to southeast into three regions (namely western IGP, central IGP, and eastern IGP), and the study period was divided into the pre-monsoon season (June) and the monsoon season (July to August). Sensitivity tests were conducted for the three regions under the two seasons."

(Page 6, Line 182-186) "As shown in Figure 3c, the sensitivity of $NH_3$ to temperature varies in different seasons and regions. Temporally, the sensitivity of $NH_3$ to temperature during the monsoon season is generally higher than that during the pre-monsoon season. Spatially, the sensitivity of $NH_3$ to temperature is highest over the eastern IGP, followed by the central IGP and the western IGP. The difference in the sensitivity of the $NH_3$ to temperature may be caused by the difference of the initial $\varepsilon(NH_4^+)$ and temperature."

[Figure]

Figure 3. Columnar $\varepsilon(NH_4^+)$ and changes of $NH_3$ total columns with the changes of temperatures predicted by ISORROPIA-II. (a) Mean columnar $\varepsilon(NH_4^+)$ and changes of $NH_3$ total columns over the IGP from June to August 2010. (b) Columnar $\varepsilon(NH_4^+)$ and (c) changes of $NH_3$ total columns over the western IGP during Pre-monsoon (PM-W), the central IGP during Pre-monsoon (PM-C), the eastern IGP during pre-monsoon (PM-E), the western IGP during monsoon (M-W), the central IGP during monsoon (M-C), the eastern IGP during monsoon (M-E).

*Referee: 4. Use of the ISORROPIA module to access the impact of emissions and temperature*

*Line 77: A more detailed overview of the ISORROPIA module would provide important context for the resulting analysis. If only applied in an offline context, it is possible that the analysis is not capturing various important (and non-linear) effects due to the spatial heterogeneity in gas phase and particle concentrations (along with the associated depositional losses at every timestep). While the authors provide an observational constraint (Line 142) to validate this approach, the differences in the model and observed partitioning ratio are significant (on the order of 30%). A more thorough comparison with the observational data would greatly benefit the analysis and serve as validation for some of the later conclusions. Additionally, the comparison of the regional mean to observational data over Delhi may not be appropriate, particularly given that NOx and SO2 concentrations are likely much higher over the city.*

**Response:** In WRF-Chem, factors that affect $NH_3$ concentrations (including transport, dry deposition, wet deposition and gas-particle portioning, etc.) are all calculated online. The MOSAIC (Model for Simulating Aerosol Interactions and Chemistry) aerosol module embedded in WRF-Chem is used for online calculation of aerosol partitioning.

As you suggested, we calculated the simulated $\varepsilon(NH_4^+)$ (partitioning ratios of $NH_4^+$ to total ammonia (TA, TA = $NH_3$ + $NH_4^+$)) during monsoon over Delhi and compared it with observational data. The simulated $\varepsilon(NH_4^+)$ during monsoon over Delhi was 0.38, which is comparable to the observed $\varepsilon(NH_4^+)$ (0.39 in the 2011 monsoon season in Delhi (Singh and Kulshrestha, 2012).

**Revision:** (Page 6, Line 163-164) "The simulated $\varepsilon(NH_4^+)$ in the 2010 monsoon in Delhi was 0.38, which is close to the observed $\varepsilon(NH_4^+)$ (0.39 in the 2011 monsoon season in Delhi) (Singh and Kulshrestha, 2012)."

*Line 155: The temperature sensitivity is an interesting result, particularly when contrasted to the SO2/NOx sensitivity. However, the approach here considers only a simplified case over the entire region. If the partitioning was conducted online (at every timestep), would it be reasonable to expect a different sensitivity to changes in temperature / SO2 / NOx? A more detailed discussion about the non-linear, spatially dependent factors driving aerosol concentrations (and the heterogeneity in emissions, loss processes, thermodynamics, etc.) would provide more context with which to interpret these results.*

**Response:** The MOSAIC (Model for Simulating Aerosol Interactions and Chemistry) aerosol module embedded in WRF-Chem is used for online calculation of aerosol partitioning. Thus, the sensitivity of $NH_3$ to changes in $SO_2/NO_x$ is the result of an online calculation.

Since WRF-Chem is a coupled model, variables such as air temperature cannot be arbitrarily perturbed. Thus, ISORROPIA-II is used offline to study the effect of air temperature on the gas-particle partitioning of $NH_3$.

Given the temporal and spatial heterogeneity, we divided the IGP from northwest to southeast into three areas (namely western IGP, central IGP, and eastern IGP), and divided the study period into two seasons (pre-monsoon and monsoon). The sensitivity of $NH_3$ to changes of temperature was analyzed for the three regions under the two seasons, see reply 5 for detail.

***Referee: 5. A discussion of other drivers of aerosol formation (particularly in the context of the monsoons)***

*Singh and Kulshrestha (2012), cited in this study, hypothesized that humidity during the monsoon season had a significant impact on NH3 partitioning. Could the authors discuss this in the context of their analysis? Given that the aim of this study is to establish the most salient drivers of the high NH3 concentrations (particularly during the selected monsoon period), a more detailed discussion of what determines the relative dominance of the different production, loss and partitioning mechanisms under various atmospheric conditions would provide important context with which to interpret the results of this analysis.*

__Response:__ Accepted. Singh and Kulshrestha (2012) found that the percent fraction of $NH_4^+$ over Delhi during monsoon was noticeably higher than that during the pre-monsoon and post-monsoon. Here, the model simulation also shows similar results. The simulated surface $\varepsilon(NH_4^+)$ over the western, central and eastern part of the IGP were 0.11, 0.13 and 0.24 during pre-monsoon and 0.26, 0.26 0.37 during monsoon. It is not difficult to find that the surface $\varepsilon(NH_4^+)$ during the monsoon season is significantly higher than that during the pre-monsoon season, and the surface $\varepsilon(NH_4^+)$ generally increases from northwest to southeast along the IGP. The spatiotemporal variations of $\varepsilon(NH_4^+)$ are consistent with the spatiotemporal variations of RH, indicating that RH is an important factor affecting the $NH_3$ partitioning.

However, it is difficult to distinguish the contribution of various factors such as emission, chemical conversion and transport. As far as we understand, the high $NH_3$ loading over the IGP appears to be the joint result of high $NH_3$ emissions, weak chemical loss, and weak horizontal diffusion. Intensive agricultural activities resulted in relatively high $NH_3$ emissions over the IGP. The low $SO_2$ and $NO_x$ emissions and high temperature limited the gas-to-particle partitioning of $NH_3$. In addition, orographic and meteorological conditions is conducive to the accumulation of $NH_3$ over the IGP.

__Revision:__ (Page 8, Line 225-231) "The simulated surface $\varepsilon(NH_4^+)$ over the western, central and eastern part of the IGP were 0.11, 0.13 and 0.24 during pre-monsoon and 0.26, 0.26 0.37 during monsoon. It is not difficult to find that the surface $\varepsilon(NH_4^+)$ during the monsoon season is significantly higher than that during the pre-monsoon season, and the surface $\varepsilon(NH_4^+)$ generally

increases from northwest to southeast along the IGP. Besides, the columnar $\varepsilon(NH_4^+)$ shows similar spatiotemporal variations with the surface $\varepsilon(NH_4^+)$ (Figure 3b). The spatiotemporal variations of $\varepsilon(NH_4^+)$ are consistent with the spatiotemporal variations of RH (Figure S5a), indicating that RH is an important factor affecting the $NH_3$ partitioning."

**References**

Binkowski, F. S., and Shankar, U.: The Regional Particulate Matter Model: 1. Model description and preliminary results, J. Geophys. Res. Atmos., 100, 26191-26209, https://doi.org/10.1029/95JD02093, 1995.

Clarisse, L., Clerbaux, C., Dentener, F., Hurtmans, D., and Coheur, P.-F.: Global ammonia distribution derived from infrared satellite observations, Nat Geosci, 2, 479-483, https://doi.org/10.1038/ngeo551, 2009.

Easter, R. C., Ghan, S. J., Zhang, Y., Saylor, R. D., Chapman, E. G., Laulainen, N. S., Abdul-Razzak, H., Leung, L. R., Bian, X., and Zaveri, R. A.: MIRAGE: Model description and evaluation of aerosols and trace gases, J. Geophys. Res. Atmos., 109, https://doi.org/10.1029/2004JD004571, 2004.

Singh, S., and Kulshrestha, U. C.: Abundance and distribution of gaseous ammonia and particulate ammonium at Delhi, India, Biogeosciences, 9, 5023-5029, https://doi.org/10.5194/bg-9-5023-2012, 2012.

Wesely, M.: Parameterization of surface resistances to gaseous dry deposition in regional-scale numerical models, Atmos. Environ., 23, 1293-1304, 1989.

Zaveri, R. A., and Peters, L. K.: A new lumped structure photochemical mechanism for large-scale applications, J. Geophys. Res. Atmos., 104, 30387-30415, https://doi.org/10.1029/1999JD900876, 1999.

Zaveri, R. A., Easter, R. C., Fast, J. D., and Peters, L. K.: Model for Simulating Aerosol Interactions and Chemistry (MOSAIC), J. Geophys. Res. Atmos., 113, D13204, https://doi.org/10.1029/2007JD008782, 2008.

**Response to Referee #2**

*Referee: The study analyses the possible factors responsible for elevated levels of ammonia over Indo-Gangetic Plain (IGP) regions during summertime. From previous studies, IGP has been identified as a hotspot for ammonia and analyzing the reasons for the high levels is an important study. Considering the implications of high-level ammonia, this study improves the understanding of the scientific community working on the ammonia over IGP. The authors use the air quality model (WRF-Chem), emission inventory and observation data (IASI satellite) to analyze the possible factors namely chemical conversion, emissions, and meteorology on ammonia loading.*
*Specific comments with line numbers are provided below and I recommend these issues to be addressed before publication in ACP.*

**Response:** We would like to thank the referrer for your detailed and constructive comments. Please see our point-by-point reply below.

*Referee: 1. Title*

*The authors classify June to August as the summer period in the study. However, the Indian summertime is considered from April-June and monsoon begins from mid-June/late June. Summer to monsoon season has a drastic change in regional meteorology over IGP and possibility impact the levels of ammonia (gas phase) over IGP from high to low because of washout effects? I would suggest the authors not to mention the summer season in the title or in the study in general if possible or provide some explanation here.*

**Response:** Accepted. As you suggested, we have corrected "summer" to "pre-monsoon and monsoon seasons". The title and the relevant part of main text have been revised.

*Referee: 2. Methods*

*Line 65: The authors mentioned the fair agreement of IASI ammonia observations with ground-based measurements citing few studies which are mainly satellite observations. It is not clear whether the fair agreement was for IGP or other regions. It would be helpful if the authors could provide some details about the ground measurements used here.*

**Response:** Accepted. The two articles cited here are validations of IASI $NH_3$ measurements using ground-based and airborne data sets around the world. In Dammers et al. (2016), IASI-$NH_3$ measurements were evaluated with ground-based Fourier transform infrared spectroscopy (FTIR) measurements from nine Network for the Detection of Atmospheric Composition Change (NDACC) stations around the world between 2008 and 2015, and results showed that IASI $NH_3$ were in fair agreement with ground-based measurements with a mean relative difference of $-32.4 \pm 56.3\%$ and a correlation r of 0.8. In Van Damme et al. (2015), IASI-$NH_3$

measurements were validated using existing independent ground-based and airborne data sets in Europe, China and Africa, results showed that IASI $NH_3$ were generally consistent with the available data sets. However, due to the lack of publicly available ammonia observation data sets in the IGP, previous studies have not evaluated IASI $NH_3$ in the IGP. For the same reason, the evaluation of IASI $NH_3$ in the IGP cannot be implemented in this study. As you suggested, we reworded in the revised manuscript.

**Revision:** (Page 3, Line 72-75) "It has been found that the IASI samples at the overpass time could represent the entire day, and IASI $NH_3$ observations are in fair agreement with the available ground-based and airborne data sets around the world (Dammers et al., 2016;Van Damme et al., 2015). However, due to the lack of publicly available ammonia observation data sets in the IGP, previous studies have not evaluated IASI $NH_3$ in the IGP."

*Line 77: The authors should provide more details on the input of the ISOROPIA-II model or in general the model in order to interpret the gas-particle partitioning of the ammonia for example what are the gas species in input?*

**Response:** Accepted. As you suggested, we added more details about the inputs of ISORROPIA-II in Sect 2.3.

**Revision:** (Page 3, Line 92-94) "The chemical and meteorological data from WRF-Chem, including water-soluble ions ($SO_4^{2-}$, $NO_3^-$, $NH_4^+$, $Cl^-$, $Na^+$) in $PM_{2.5}$, gaseous precursors ($NH_3$, $HNO_3$, $HCl$), temperature (T) and relative humidity (RH) are used as the inputs of ISORROPIA-II."

*Line 80: "As inputs of ISORROPIA-II, the outputs (water-soluble ions, gas species, T and RH) of WRF-Chem were first averaged over the IGP and then averaged for summer 2010". Please provide more details on averaging here. As IGP is a vast region, averaging the inputs over the region may create a bias for some regions over IGP considering the heterogeneity of the sources over IGP. Again the time averaging may create some bias too as the time period included in the study has few days or a month of the summer season and two months of monsoon as per Indian meteorological department classification. It would be useful to check the variability of the inputs month wise for example June, July, and August separately.*

**Response:** Accepted. Previously, the offline calculation of ISORROPIA-II was based on the average of the entire IGP from June to August, which might cause bias due to the spatial and temporal heterogeneity. As you suggested, we divided the IGP from northwest to southeast into three areas (namely western IGP, central IGP, and eastern IGP), and divided the study period into two seasons (pre-monsoon and monsoon), and analyzed the sensitivity of $NH_3$ to temperature for the three regions under the two seasons in Sect 3.2.

**Revision:** (Page 4, Line 98-101) "Sensitivity tests was firstly conducted based on the average of the entire IGP from June to August. Then, the IGP was divided equally from northwest to southeast into three regions (namely western IGP, central IGP, and eastern IGP), and the study period was divided into the pre-monsoon season (June) and the monsoon season (July to August). Sensitivity tests were conducted for the three regions under the two seasons."

(Page 6, Line 182-186) "As shown in Figure 3c, the sensitivity of $NH_3$ to temperature varies in different seasons and regions. Temporally, the sensitivity of $NH_3$ to temperature during the monsoon season is generally higher than that during the pre-monsoon season. Spatially, the sensitivity of $NH_3$ to temperature is highest over the eastern IGP, followed by the central IGP and the western IGP. The difference in the sensitivity of the $NH_3$ to temperature may be caused by the difference of the initial $\varepsilon(NH_4^+)$ and temperature."

[Figure]

Figure 3. Columnar $\varepsilon(NH_4^+)$ and changes of $NH_3$ total columns with the changes of temperatures predicted by ISORROPIA-II. (a) Mean columnar $\varepsilon(NH_4^+)$ and changes of $NH_3$ total columns over the IGP from June to August 2010. (b) Columnar $\varepsilon(NH_4^+)$ and (c) changes of NH3 total columns over the western IGP during Pre-monsoon (PM-W), the central IGP during Pre-monsoon (PM-C), the eastern IGP during pre-monsoon (PM-E), the western IGP during monsoon (M-W), the central IGP during monsoon (M-C), the eastern IGP during monsoon (M-E).

*Referee: 3. Results*

*Line 105: As per the sowing season, IGP has mainly two cropping cycles which includes summer and winter both. During both cycles, fertilizer applications can be intensive. Is there any study/data supporting the highest N- fertilizer application during the months mentioned in this study? Line108: Authors cited the Riddick et al. 2010 for the exponential increase of ammonia emission with temperature. Would it be possible to expect similar results, if the temperature data and ammonia satellite observations from the present study taken in to account?*

**Response:** The monthly map of N-fertilizer application amounts from Nishina et al. (2017) shows that there two peaks in N-fertilizer application amounts over the IGP, one in May-August, the other in November-December, which is consistent with the two cropping cycles in the IGP. We reworded in the

revised manuscript.

However, it is difficult to study the relationship between $NH_3$ emission and temperature in this study. Because $NH_3$ satellite observations are affected by many factors, and are not linearly related with $NH_3$ emissions. Thus, it is difficult to isolate the influence of temperature using $NH_3$ observations.

**Revision:** (Page 4, Line 121-125) "IGP has two cropping cycles including summer and winter (2012). June to August is one of the two main sowing periods in the IGP with a large amount of N-fertilizer applied to the cropland as base fertilizer. The monthly map of N-fertilizer application amounts from Nishina et al. (2017) shows that there are two peaks in N-fertilizer application amounts over the IGP with one in May-August, the other in November-December, which is consistent with the two cropping cycles in the IGP."

*Line 160: The analysis of the low gas-to-particle conversion of ammonia demonstrates the sensitivity to SO2/NOx emissions. This is an excellent analysis based on the model output data. Is it fair to expect similar results if the data for SO2/NOx not modeled but taken from observations either ground or satellite observations? Please comment on this.*

**Response:** Accepted. The spatial distributions of satellite-derived $SO_2$ and $NO_2$ columns, as shown in Figure S2, further demonstrate the lower $SO_2$ and $NO_2$ concentrations over the IGP compared to the NCP. The $SO_2$ and $NO_2$ columns over the IGP (0.5 and $2.3 \times 10^{15}$ molecules $cm^{-2}$) are much lower than that over the NCP (10.4 and $8.3 \times 10^{15}$ molecules $cm^{-2}$).

**Revision:** (Page 3, Line 78-81) "$SO_2$ columns from June to August 2010 were derived from the Level-3 Aura/OMI Global $SO_2$ Data Products (OMSO2e) (Krotkov et al., 2015). Tropospheric $NO_2$ columns from Ozone Monitoring Instrument (OMI) aboard NASA Aura satellite were used from June to August 2010 (http://www.temis.nl/airpollution/no2col/no2regioomimonth_qa.php)"

(Page 5, Line 136-138) "Besides, the satellite-derived $SO_2$ and $NO_2$ columns over the IGP (0.5 and $2.3 \times 10^{15}$ molecules $cm^{-2}$) are also much lower than that over the NCP (10.4 and $8.3 \times 10^{15}$ molecules $cm^{-2}$) (Fig. S2)."

[Figure]

Figure S2. The spatial distribution of (a) SO$_2$ and (b) NO$_2$ columns over East Asia from June to August 2010.

*Line 200: The authors concluded an interesting observation about the other factors than wet conditions and high RH controlling the high ammonia loading over the IGP. More details on this would be useful to support this conclusion. Even a simple time series of IASI ammonia observations with RH for the period of study would provide an important context.*

**Response:** Accepted. As you suggested, we added more details about the effect of RH on gas-to-particle partitioning of NH$_3$ in Sect 3.3.

**Revision:** (Page 8, Line 225-231) "The simulated surface $\varepsilon(NH_4^+)$ over the western, central and eastern part of the IGP were 0.11, 0.13 and 0.24 during pre-monsoon and 0.26, 0.26 0.37 during monsoon. It is not difficult to find that the surface $\varepsilon(NH_4^+)$ during the monsoon season is significantly higher than that during the pre-monsoon season, and the surface $\varepsilon(NH_4^+)$ generally increases from northwest to southeast along the IGP. Besides, the columnar $\varepsilon(NH_4^+)$ shows similar spatiotemporal variations with the surface $\varepsilon(NH_4^+)$ (Figure 3b). The spatiotemporal variations of $\varepsilon(NH_4^+)$ are consistent with the spatiotemporal variations of RH (Figure S5a), indicating that RH is an important factor affecting the NH$_3$ partitioning."

**References**

Agricultural Statistics At a Glance 2012, Department of Agriculture and Cooperation, Ministry of Agriculture, Government of India, 2012.

Dammers, E., Palm, M., Van Damme, M., Vigouroux, C., Smale, D., Conway, S., Toon, G. C., Jones, N., Nussbaumer, E., Warneke, T., Petri, C., Clarisse, L., Clerbaux, C., Hermans, C., Lutsch, E., Strong, K., Hannigan, J. W., Nakajima, H., Morino, I., Herrera, B., Stremme, W., Grutter, M., Schaap, M., Wichink Kruit, R. J., Notholt, J., Coheur, P.-F., and Erisman, J. W.: An evaluation of IASI-NH3 with ground-based Fourier transform infrared spectroscopy measurements, Atmos Chem Phys, 16, 10351-10368, https://doi.org/10.5194/acp-16-10351-2016, 2016.

Krotkov, N. A., Li, C., and Leonard, P.: OMI/Aura Sulfur Dioxide (SO$_2$) Total Column L3 1 day Best

Pixel in 0.25 degree x 0.25 degree V3, Goddard Earth Sciences Data and Information Services Center (GES DISC), https://doi.org/10.5067/Aura/OMI/DATA3008, 2015.

Nishina, K., Ito, A., Hanasaki, N., and Hayashi, S.: Reconstruction of spatially detailed global map of NH4+ and NO3- application in synthetic nitrogen fertilizer, Earth Syst Sci Data, 9, 149-162, https://doi.org/10.5194/essd-9-149-2017, 2017.

Van Damme, M., Clarisse, L., Dammers, E., Liu, X., Nowak, J. B., Clerbaux, C., Flechard, C. R., Galy-Lacaux, C., Xu, W., Neuman, J. A., Tang, Y. S., Sutton, M. A., Erisman, J. W., and Coheur, P. F.: Towards validation of ammonia (NH3) measurements from the IASI satellite, Atmospheric Measurement Techniques, 8, 1575-1591, https://doi.org/10.5194/amt-8-1575-2015, 2015.

---

## Author Response (AR2)

Our point-by-point responses are provided below. The comments are italicized.

**Response to Editor**

*Editor: Many thanks for your careful revision. Please note that Referee #1 has a few more comments that should be addressed in a further revision. I also added a few minor/technical comments below that should be considered. When all of these comments are addressed satisfactorily, I will be happy to accept your manuscript for publication in ACP.*

**Response:** We would like to thank the editor for your detailed and constructive comments. Please see our point-by-point reply below.

*Editor: l. 44: remove or specify 'all possible factors'*

**Response:** Accepted. We removed 'all possible factors' at line 44.

*Editor: l. 57-64: It seems that this text may have been copied form another source as the reference formatting is not according to ACP guidelines. 1) Reword the text if it has been used previously, 2) Add correct references instead of numbers (1) – (5)*

**Response:** Accepted. Revised at line 57-63.

**Revision:** (Page 2, Line 57-63) "CBM-Z (Carbon Bond Mechanism version Z) chemical mechanism (Zaveri and Peters, 1999) and MOSAIC (Model for Simulating Aerosol Interactions and Chemistry) aerosol module (Zaveri et al., 2008) were used for modeling gas phase photochemistry and aerosol processes (including nucleation, coagulation, condensation and thermodynamic equilibrium), respectively. Dry deposition for trace gases and aerosols was treated following the methods of Wesely (1989) and Binkowski and Shankar (1995), respectively. Wet deposition in the model includes both in-cloud and below-cloud scavenging. The below-cloud scavenging of aerosols and trace gases was calculated based on the methods of Easter et al. (2004)."

*Editor: l. 61: 'below-cloud scavenging'*

**Response:** Accepted. Revised at line 62.

*Editor: Section 2.3: 1) How was the presence of organics and their related water uptake accounted for? I assume that they are included in ISORROPIA; however, this and the possible bias should be mentioned; 2) What is the reasoning for the large variation of air temperature of 20C, i.e. +/- to measured(?) temperature? Was this variation in the model based on measured temperature variability?*

**Response:** Organic species are not considered in the thermodynamic calculations in

ISORROPIA-II, because the impact of organic species on aerosol thermodynamics is still rather poorly understood (Fountoukis and Nenes, 2007). Pye et al. (2018) found that the AIOMFAC (Aerosol Inorganic–Organic Mixtures Functional groups Activity Coefficients) based equilibrium model considering inorganic-organic interactions was consistent with ISORROPIA in terms of $NH_3$ gas-particle partitioning. Metzger et al. (2006) found that the ammonium partitioning ratio (the concentration ratio of ammonium to the total of ammonia and ammonium) calculated by ISORROPIA was about 15% lower than that calculated by EQSAM2 (Equilibrium Simplified Aerosol Model) considering organic acids. Thus, the influence of organic species on the $NH_3$ gas-particle partitioning might be limited and will not have a significant impact on the results of this study. However, these two studies were conducted in the United States. The effects of organic species on aerosol thermodynamics in the IGP need further research in the future. As you suggested, we discussed the possible bias related to organics in section 4.

The variation of air temperature was not based on measured temperature variability. We just selected a sufficiently large variation range of air temperature to investigate the sensitivity of $NH_3$ total columns to air temperature.

**Revision:** (Page 9, Line 256-264) "Besides, organic species are not considered in the thermodynamic calculations in this study, because the impact of organic species on aerosol thermodynamics is still rather poorly understood (Zaveri et al., 2008;Fountoukis and Nenes, 2007)). Pye et al. (2018) found that the AIOMFAC (Aerosol Inorganic–Organic Mixtures Functional groups Activity Coefficients) based equilibrium model considering inorganic-organic interactions was consistent with ISORROPIA in terms of $NH_3$ gas-particle partitioning. Metzger et al. (2006) found that the $\varepsilon(NH_4^+)$ calculated by ISORROPIA was about 15% lower than that calculated by EQSAM2 (Equilibrium Simplified Aerosol Model) considering organic acids. Thus, the influence of organic species on the $NH_3$ gas-particle partitioning might be limited and will not have a significant impact on the results of this study. However, these two studies were conducted in the United States. The effects of organic species on aerosol thermodynamics in the IGP need further research in the future."

***Editor:** l. 98: replace 'was firstly' by 'were firstly'*

**Response:** Accepted. Revised at line 99.

***Editor:** l. 105: replace 'was 9.9 Tg' by 'were 9.9 Tg'*

**Response:** Accepted. Revised at line 106.

*Editor: l. 116; 117; 121: References seem incomplete (2012a) (2012b)*

**Response:** Accepted. Revised at line 117,118 and 122.

*Editor: l. 141: 'richness' seems unusual word here. Do you mean 'excess' ?*

**Response:** Accepted. We replaced 'richness' by 'excess' at line 142.

*Editor: l. 217: spell out ABL*

**Response:** Accepted. Revised at line 216 and 218.

**Response to Referee #1**

*Referee: This study aims to explore the reasons behind the elevated levels of ammonia observed over the Indo-Gangetic Plain (IGP). This is an important and scientifically relevant question, particularly since the ammonia burden has significant implications on inorganic aerosol concentrations over the region. The authors use the WRF-Chem model to investigate the physics and thermodynamics underlying the atmospheric fate of NH3 and the resulting analysis provides useful insights into some of the factors driving the high concentrations over the region. Specifically, the authors suggest that despite other regions (such as the North China Plain) having higher ammonia emissions, the IGP has a higher burden of ammonia primarily due to:*
*1) The lower amounts of NOx and SO2 emissions over the region compared to the NCP (limiting aerosol formation)*
*2) The terrain and meteorology of the region that result in weak horizontal advection and the accumulation of NH3*

**Response:** We would like to thank the referee for your detailed and constructive comments. Please see our point-by-point reply below.

*Referee: While the physical transport and meteorology simulated by the model is validated and constrained, the ammonia simulation and the underlying processes are not explicitly validated. The authors contend that this is due to the lack of available data in the region. However, as a result, while the study provides useful sensitivity analyses, the conclusions are largely model-driven estimations. With this in mind, the study could be strengthened by a more detailed discussion of the underlying uncertainties associated with the different model processes that drive the pathways that the authors identify as being most important to the elevated ammonia burden. The authors could also add a more specific discussion about future work that might observationally validate these conclusions and improve model NH3 representation over this region.*

**Response:** Accepted. We discussed the uncertainties associated with the different model processes and the observation work needed in the future in section 4.

**Revision:** (Page 9, Line 253-266) "The gas-particle partitioning plays an important role in influencing NH$_3$ columns. The deviation of the simulated sulfate and nitrate will cause a deviation in the simulated NH$_3$ by affecting NH$_3$ gas-particle partitioning. Thus, in addition to the NH$_3$ and NH$_4^+$, the simulated concentrations of sulfate and nitrate are also necessary to be constrained using field observations in the future. Besides, organic species are not considered in the thermodynamic calculations in this study, because the impact of organic species on aerosol thermodynamics is still rather poorly understood (Zaveri et al., 2008;Fountoukis and Nenes, 2007)). Pye et al. (2018) found that the AIOMFAC (Aerosol Inorganic–Organic Mixtures Functional groups Activity Coefficients) based equilibrium model considering inorganic-organic interactions was consistent with

ISORROPIA in terms of NH$_3$ gas-particle partitioning. Metzger et al. (2006) found that the ε(NH$_4^+$) calculated by ISORROPIA was about 15% lower than that calculated by EQSAM2 (Equilibrium Simplified Aerosol Model) considering organic acids. Thus, the influence of organic species on the NH$_3$ gas-particle partitioning might be limited and will not have a significant impact on the results of this study. However, these two studies were conducted in the United States. The effects of organic species on aerosol thermodynamics in the IGP need further research in the future. Additionally, dry and wet deposition also has an important influence on NH$_3$ columns. Field observations of the dry and wet deposition of NH$_3$ and NH$_4^+$ in the IGP are needed to constrain model simulations in the future."

*Referee: I also suggest the removal of the statement on line 44 - "This is the first study to analyze the causes of high NH3 loading over the IGP considering all possible factors." – since the study does not exhaustively evaluate various important processes (such as deposition, regional atmospheric chemistry, etc.) that could play an important role.*

**Response:** Accepted. We removed 'all possible factors' at line 44.

**References**

[revised manuscript text omitted]
 from June to August 2010 estimated using MIX database. Figure S2 shows the spatial distributions of $SO_2$ and $NO_2$ columns over East Asia. Figure S3 provides the comparison of the simulated $SO_4^{2-}$ and $NO_3^-$ concentrations in the base case and the increased emissions case. Figure S4 shows the spatial distribution of the $NH_3$ total columns from June to August 2010 derived from IASI measurements. Figure S5 shows spatial distributions of WRF-Chem predicted relative humidity and precipitation from June to August 2010, and the circles in Figure S5a represent the observed relative humidity obtained from NCDC dataset. Table S1 lists the options of WRF-Chem configurations. Table S2 provides the performance statistics of meteorological predictions of WRF-Chem.

[Figure]

**Figure S1.** Spatial distributions of emission fluxes of (a) NH$_3$, (b) SO$_2$, and (c) NO$_x$ over East Asia from June to August 2010. The blue quadrangle represents the IGP, and the green quadrangle represents the NCP.

[Figure]

**Figure S2.** The spatial distributions of (a) SO₂ and (b) NO₂ columns over East Asia from June to August 2010.

[Figure]

**Figure S3.** Spatial distributions of WRF-Chem predicted SO₄²⁻ and NO₃⁻ concentrations from June to August 2010. (a) and (b) are SO₄²⁻ concentrations in the base case and the increased emissions case, respectively. (c) and (d) are NO₃⁻ concentrations in the base case and the increased emissions case, respectively.

[Figure]

**Figure S4.** The spatial distribution of NH₃ total columns from June to August 2010 retrieved from IASI measurements.

[Figure]

**Figure S5.** Spatial distributions of WRF-Chem predicted meteorological variables from June to August 2010. (a) Relative humidity. (b) Precipitation. Circles in (a) show the observed Relative humidity.

**Table S1.** WRF-Chem configurations

| | |
|---|---|
| Meteorology initial and boundary conditions | Reanalysis data from the National Centers for Environmental Prediction Final Analysis (NCEP-FNL) |
| Shortwave radiation | rapid radiative transfer model (RRTMG) |
| Longwave radiation | rapid radiative transfer model (RRTMG) |
| Land surface model | Noah land-surface model |
| Planetary boundary layer model | Mellor-Yamada-Janjic (Eta) TKE scheme |
| Cumulus parameterization | New Grell scheme (G3) |
| Microphysics | Lin et al. Scheme |
| Photolysis | Fast-J photolysis |

**Table S2.** Performance statistics of meteorological predictions of WRF-Chem.

| | T2[a] | RH2[a] | WS10[a] | WD10[a] |
|---|---|---|---|---|
| Data pairs[b] | 27508 | 27443 | 18036 | 18036 |
| MeanObs[b] | 30.9 | 69.3 | 2.7 | 165.1 |
| MeanSim[b] | 31.9 | 63.3 | 2.9 | 152.9 |
| R[b] | 0.8 | 0.8 | 0.1 | 0.4 |
| MB[b] | 0.9 | -5.9 | 0.2 | -12.1 |
| RMSE[b] | 3.6 | 19.8 | 2.7 | 95.2 |
| NMB (%)[b] | 3.0 | -8.6 | 9.1 | -7.4 |

[a] T2: temperature at 2 m; RH2: relative humidity at 2 m; WS10: wind speed at 10 m; WD10: wind direction at 10 m; SLP: sea level pressure.

[b] data pairs: the number of observed and simulated data pairs; MeanObs: mean observational data; MeanSim: mean simulation results; R: correlation coefficient; MB: mean bias; RMSE: root mean square error; NMB: normalized mean bias.